# A Honeycomb-like Ammonium-Ion Fiber Battery with High and Stable Performance for Wearable Energy Storage

**DOI:** 10.3390/polym14194149

**Published:** 2022-10-03

**Authors:** Jiangdong Sun, Wenqi Nie, Shuai Xu, Pengxiang Gao, Shuang Sun, Xianhong Zheng, Qiaole Hu, Zhenzhen Xu

**Affiliations:** 1School of Textile and Garment, Anhui Polytechnic University, Wuhu 241000, China; 2Key Laboratory of Bio-Fibers and Eco-Textiles, Qingdao University, Qingdao 266071, China; 3Key Laboratory of Eco-Textiles, Ministry of Education, Jiangnan University, Wuxi 214122, China

**Keywords:** ammonium-ion batteries, honeycomb-like, long cycle-life, aqueous, flexible

## Abstract

Aqueous ammonium-ion batteries have attracted intense interest lately as promising energy storage systems due to the price advantage and fast charge/discharge capability of ammonium-ion redox reactions. However, the research on the strength and energy storage characteristics of ammonium-ion fiber batteries is still limited. In this study, an ammonium-ion fiber battery with excellent mechanical strength, flexibility, high specific capacity, and long cycle-life has been developed with a robust honeycomb-like ammonium vanadate@carbon nanotube (NH_4_V_4_O_10_@CNT) cathode. The fiber electrode delivers a steady specific capacity of 241.06 mAh cm^−3^ at a current of 0.2 mA. Moreover, a fiber full cell consisting of an NH_4_V_4_O_10_@CNT cathode and a PANI@CNT anode exhibits a specific capacity of 7.27 mAh cm^−3^ at a current of 0.3 mA and retains a high capacity retention of 72.1% after 1000 cycles. Meanwhile, it shows good flexibility and superior electrochemical performance after 500 times bending or at different deformation states. This work offers a reference for long-cycle, flexible fibrous ammonium-ion batteries.

## 1. Introduction

The booming of the Fourth Industrial Revolution promotes the development of artificial intelligence, 5G information, big data, distributed mobile internet, etc., in society [1,2,3]. Micro energy sources are increasingly supplied for new electronic devices. Exploring high-energy-density, lightweight, and safely used rechargeable batteries has attracted great attention recently. Compared with traditional batteries, aqueous batteries are considered a crucial component of next-generation energy storage due to their high level of safety and environmental friendliness. In the past five years, great progress has been made in the research on aqueous batteries, including Li-ion batteries, Zn-ion batteries, K-ion batteries, and Na-ion batteries [4,5,6,7]. Notably, the redox reaction of the metal ions serves as the foundation for the operation of these aqueous batteries. Although metal-ion battery systems have made a certain breakthrough in nanomaterial synthesis, electrode preparation, and solid electrolytes, it is still a challenge to develop appropriate metal-ion electrode systems with resourceful, fast ion diffusion and high energy density [8,9,10,11,12,13,14].

Recently, non-metal-ion aqueous batteries have been proposed as new candidate energy systems due to their abundant resources, large interlayer spacing, and fast diffusion capability of non-metal carriers [15,16,17]. NH_4_-ion batteries, in particular, with a molar mass of 18 g mol^−1^ and a charge carrier of hydration ion size, have the potential to replace metal-ion aqueous batteries [18,19,20]. Much more effort has been made to improve the performance of NH_4_-ion batteries. Song et al. [21] reported an electrodeposited manganese oxide (MnO_x_) cathode. Carbon cloth was chosen as the collector, and the electrochemical test results showed that the layered structure of MnO_x_ had great rate capability with a specific capacity of 176 mAh g^−1^ at a current density of 0.5 A g^−1^, and 66 mAh g^−1^ even at a high rate of 10 A g^−1^. In addition, the capacity retention rate is 80% after 600 cycles at 1 A g^−1^, indicating excellent cyclic stability. Besides manganese oxides, Prussian Blue analog (PBA) materials were also worthy of attention. Wu et al. [22] successfully prepared a new PBA cathode (Ni-APW). The Ni-APW cathode delivered a high capacity of 40 mAh g^−1^ at 10 C; even at high current densities of 10 C and 20 C, the specific capacities reached 28 and 20 mAh g^−1^. Impressively, when the current density is 5 C, the capacity retention rate is up to 74% after 2000 cycles. Furthermore, an aqueous NH_4_-ion full cell was constructed with Ni-APW as the cathode, 3,4,9,10-perylene tetracarboxylic diimide (PTCDI) as the electrode, and 1.0 M (NH_4_)_2_SO_4_ solution as the electrolyte. The full cell demonstrated good performance with an average operation voltage of 1.0 V, a capacity retention of 67% after 1000 cycles, and an energy density of 43 Wh kg^−1^. Despite these efforts, the research on NH_4_-ion full cells is still in its early stages, and practical application is far from satisfactory. 

Meanwhile, to address the demand for energy supply in the flexible wearable electronic field, significant research effort should be aimed at the development of high-performance and high-safety flexible aqueous batteries. However, it is a pity that there are few reports of flexible NH_4_-ion batteries so far. Li et al. [23] reported the first fiber-shaped aqueous full-cell NH_4_-ion battery. Carbon fiber was chosen as a current collector, and the NH_4_-ion fiber electrode was prepared by a simple hydrothermal method. The result showed the fiber-shaped battery had excellent electrochemical performance with a specific capacity of 167 mAh g^−1^. Although the specific capacitance was lower than that of metal-ion batteries, it opened up new research fields and expanded the types of battery materials. However, the cycle stability of NH4-ion fiber is not undesirable, with 73% retention of initial capacitance after 1000 cycles. Wang [24] also reported a full flexible ammonium-ion battery consisting of a concentrated hydrogel electrolyte sandwiched between the NH_4_V_3_O_8_·2.9H_2_O nanobelt cathode and the PANI anode. Test results showed that this flexible battery provides a specific capacity of 121 mAh g^−1^, as well as a capacity retention rate of 95% even after 400 cycles. In addition, the electrochemical properties of this cell remain nearly unchanged after 90 and 180 degrees of bending, demonstrating remarkable mechanical strength and flexibility. Despite the NH_4_-ion battery research being encouraging, fiber electrodes with high specific capacity and high strength still need to be further explored.

To improve the energy storage and recycle life of the NH_4_-ion fiber battery, herein, we report an NH_4_-ion fiber battery with a high specific capacity, excellent rate capability, and outstanding long cycle-life. A honeycomb-like NH_4_V_4_O_10_@CNT electrode with large d-spacing for NH_4_-ion fast diffusion was set as the cathode, and PANI@CNT fiber worked as the anode. The electrochemical performance test result shows that the specific capacity of NH_4_V_4_O_10_@CNT is up to 241.06 mAh cm^−3^ at a current of 0.2 mA with a Coulombic efficiency of about 97.3%. Furthermore, the full fiber battery shows an outstanding specific capacity of 7.27 mAh cm^−3^, excellent cycling stability with a capacity retention of 72.1% after 1000 cycles at 0.5 mA, and good flexibility with maintained energy at different deformation states; it can be integrated with textile and exhibits great potential application in wearable textiles. The advantages of good electrochemical performance, long recycle-life, and high flexibility make the NH_4_V_4_O_10_@CNT fiber electrode a promising candidate for energy storage used in wearable electronics (as shown in Figure 1).

## 2. Experimental Section

### 2.1. Materials

#### 2.1.1. The CNT Fiber 

The CNT fiber was produced by drawing and twisting a CNT array made by the chemical vapor deposition (CVD) method. In brief, with C_2_H_4_ as the carbon source and Ar as the carrier gas, the CNT sheet could be deposited on SiO_2_/Si substrate. When the well-aligned CNT fiber was drawn out of the CNT array, one side of the CNT array was caught and twisted by a twisting instrument to produce the CNT fiber.

#### 2.1.2. NH_4_V_4_O_10_@CNT Electrode

A foam-like NH_4_V_4_O_10_@CNT electrode was prepared by a simple hydrothermal method [23]. First, 0.282 g ammonium vanadate (NH_4_VO_3_) was dissolved in 30 mL of deionized water with magnetic stirring. After it completely dissolved, 0.34 g b-cyclodextrin (bCD) was then added to the solution and stirred for 40 min at room temperature. Then, 0.325 g oxalic acid (H_2_C_2_O_4_) was added and stirred for another 30 min. After the homogenization, the pretreated CNT fiber was immersed in the solution, which was transferred into 50 mL Teflon-lined stainless steel autoclaves for the hydrothermal reaction, and the pretreated CNT fiber was immersed in the solution. After that, the autoclave was heated at 120 ℃ for 20 h in an electric oven. After cooling, the NH_4_V_4_O_10_@CNT was collected (as shown in Figure 1). Moreover, excess NH_4_V_4_O_10_ was removed with deionized water and ethanol before drying at 60 °C for 4 h. The diameter of the NH_4_V_4_O_10_@CNT electrode was 80 μm on average.

#### 2.1.3. PANI@CNT Electrode

PANI@CNT fiber was prepared by electrochemical polymerization with a three-electrode configuration. A Pt sheet and a Ag/AgCl electrode served as the counter and reference electrodes, respectively. The electrolyte was a solution of 0.2 mol L^−1^ aniline and 0.5 mol L^−1^ H_2_SO_4_. The test method was cyclic voltammetry (CV) in a potential ranging from −0.2 to 1.2 V, the scan rate was maintained at 20 mV s^−1^, and the segment was 12 cycles. 

### 2.2. Material Characterization

Morphologies of the NH_4_V_4_O_10_ powder were characterized using a scanning electron microscope (SEM), and energy-dispersive spectroscopy (EDS) was also performed on a field-emission SEM (Hitachi, Tokyo, Japan). The X-ray diffraction (XRD) patterns were obtained using the Bruker D8 Advance X-ray diffractometer (Bruker, Karlsruhe, Germany). Transmission electron microscopy (TEM) (Hitachi, Tokyo, Japan) was also conducted. Fiber tensile tests were carried out with an Electronic Strength Tester machine (Wenzhou Fangyuan instrument Co., LTDWenzhou, China). The test method was conformed to ASTM D3397-75 at a stretching rate of 10 mm/min.

### 2.3. Electrochemical Test

Electrochemical measurements were performed using an electrochemical workstation (Autolab PGSTAT 204, Metrohm, Herisau, Switzerland) at room temperature. To test the electrochemical properties of the NH_4_V_4_O_10_@CNT fiber electrode, we used a three-electrode system. A Pt sheet served as the counter electrode, and the reference electrode was Ag/AgCl. To prevent other ions from affecting the dynamic performance of the NH_4_V_4_O_10_@CNT electrode, we used NH_4_Cl solution (about 4.5 mol L^−1^) instead of saturated KCl solution. The electrolyte was an aqueous solution of 1 mol L^−1^ (NH_4_)_2_SO_4_. The voltage window range was 0~1 V for the CV test. The electrochemical properties of the PANI@CNT fiber electrode were also tested using a three-electrode system, but the electrolyte was 1 mol L^−1^ aqueous solution of NaCl. Moreover, the voltage window of the CV test was −0.1~0.5 V. Electrochemical impedance spectroscopy (EIS) measurements were performed in a frequency range of 100 kHz to 0.01 Hz with an AC amplitude of 10 mV. For the NH_4_-ion full cell, electrochemical performance was tested using a two-electrode system. A long cycle test was conducted using a Land CT2001A battery testing system.

### 2.4. Preparation of Fiber-Shaped NH_4_-Ion Full Cell 

To facilitate electrochemical performance testing, the NH_4_V_4_O_10_@CNT electrode and PANI@CNT fiber electrode were linked with copper wires using conductive silver adhesives. PANI@CNT was entangled with glass fiber as a separate fiber to prevent a short circuit. The electrolyte was an aqueous solution of 1 mol L^−1^ (NH_4_)_2_SO_4._ Finally, the electrode and electrolyte were all added into a heat-shrinkable tube to assemble the fiber-shaped NH_4_-ion battery.

## 3. Results and Discussion

Figure 2 shows the SEM images of NH_4_V_4_O_10_ powder and NH_4_V_4_O_10_@CNT fiber. The NH_4_V_4_O_10_ powder structure was circular, resembling an urchin (Figure 2a). The magnification picture showed that this circle was composed of many rectangular rods (Figure 2b,c). Many nanorods were connected to form a three-dimensional circle structure. Using the same hydrothermal reaction conditions, the CNT fibers were set into autoclaves for 2 h until the reaction was over. It could be seen that the color of the CNT fiber surface changed from black to green. From SEM images (Figure 2d,e), it could be seen that NH_4_V_4_O_10_ nanorods were chemically deposited on the surface of the CNT fiber, resembling a honeycomb or porous foam. This kind of honeycomb structure, regularly stacked with NH_4_V_4_O_10_ nanorods (Figure 2f), formed more micropores, which was beneficial in promoting ion transfer and improving the reaction rate of the electron–ion interface of the electrode.

In order to understand the microstructure properties of synthetic NH_4_-ion powder, many microscopic test characterizations were performed. The XRD patterns of the NH_4_-ion powder are shown in Figure 3a. The XRD characteristic peaks of NH4-ion powder were well indexed to the pure NH_4_V_4_O_10_. The lattice parameters were *a* = 11.71 Å, *b* = 3.66 Å, *c* = 9.72 Å precisely, with a space group of C 2/m. A strong peak (001) was observed at 8.8°, and the large interlayer spacing (around 9.8) was associated with the presence of NH_4_^+^ between the adjacent V-O layers [25]. It revealed that the large d-spacing accommodated the aqueous ion well and facilitated its diffusion.

The transmission electron microscopy (TEM) images displayed that the nanorods exfoliated from the “honeycomb” during TEM sample preparation, and the width was about 100 nm (Figure 3c). The HRTEM picture (Figure 3d) showed that the thickness of the nanorods was 16 nm and the interlayer spacing of the structure was 0.966 nm. This result was almost consistent with the XRD data on interplanar spacing. In addition, the EDS mapping of the NH_4_V_4_O_10_@CNT fiber exhibited the distribution of C, N, V, and O (Figure 3b and Appendix A). It was observed that N, V, and O elements were rather uniformly distributed, implying that homogeneous growth of NH_4_V_4_O_10_ on the CNT fiber surfaces was achieved.

The tensile strength of the NH_4_V_4_O_10_@CNT fiber was characterized according to ASTM D2297-75 at a stretching rate of 10 mm/min, as shown in Appendix A. A high tensile strength of 300.2 cN (~100.1 cN/dtex) was obtained by NH_4_V_4_O_10_ grown on the surface of CNT fiber, higher than the CNT fiber stress of 263 cN (~87.7 cN/dtex). This means it can be integrated into the textile process and applied as a power source for flexible electronics. To better understand the energy storage behavior of the as-fabricated NH_4_^+^-ion fiber battery, the electrochemical performance of the NH_4_V_4_O_10_@CNT fiber was tested in a three-electrode system with the Autolab electrochemical workstation. Figure 4a shows the first three cyclic voltammetry (CV) curves of the NH_4_V_4_O_10_@CNT fiber in 1 mol L^−1^ (NH_4_)_2_SO_4_ aqueous solution electrolyte at a scan rate of 20 mV^−1^. One pair of oxidation peaks and reduction peaks located at approximately 0.74 and 0.59 V vs. Ag/AgCl in the first several cycles can be seen. According to a previous study, the CNT fiber is a double-layer capacitor, so there is no oxidation peak in the CV curves of electrochemical performance [26]. This means the anodic and cathodic peaks corresponded to electrode reaction kinetics of NH_4_ ion intercalation and de-intercalation from the NH_4_V_4_O_10_@CNT fiber electrode [27,28]. Incidentally, the CV curves in the first three cycles overlapped excellently, implying NH_4_^+^ de-intercalation/intercalation on the fiber electrode reversibly. Figure 4b further characterizes the CV performance of the NH_4_V_4_O_10_@CNT fiber electrode at various scan rates from 0.5 to 50 mV ^−1^. It can be observed that two pairs of redox peaks always appeared and were located almost in the same position at about 0.72/0.55 V. This was attributed to the processes of NH_4_^+^ oxidization and reduction [29]. To gain more insight, the EIS test was also conducted for the NH_4_V_4_O_10_@CNT fiber electrode. The Nyquist plot is shown in Figure 4c and the inset picture. The slope of the line was greater than 45°, implying reaction kinetics of the electrode contained both capacitor-like and battery-like types, with the capacitive feature occupying the major portion. The inset curve is the high frequency of the Nyquist plot. The semicircle in the high frequency indicates charge transfer resistance (R_ct_) between electrode and electrolyte, and the value is about 12 Ω, which is lower than the 42 Ω of NH_4_V_4_O_10_ [30], verifying that the electrode reaction kinetics speed is fast and the resistance is small.

To explore the charging and discharging results of electrodes at different current densities, the galvanostatic charge–discharge (GCD) plots of NH_4_V_4_O_10_@CNT fiber are given, as Figure 4d shows. The NH_4_V_4_O_10_@CNT fiber electrode had a high specific capacitance of 241.06 mAh cm^−3^ at a current of 0.2 mA and a Coulombic efficiency of about 97.3% at the same current, indicating high specific capacity. At the same time, as for the NH_4_V_4_O_10_@CNT fiber battery, capacitances of 215.63, 199.04, 179.14, and 168.08 mAh cm^−3^ corresponded to 0.3 mA, 0.4 mA, 0.6 mA, and 0.8 mA of current, respectively (Appendix A), showing an excellent rate capability. This was due to the fast electron/ion transport in the honeycomb structure.

Additionally, the formula current, *i(V)*, vs. scan rate *v* can be easily calculated and help us better understand the diffusion/non-diffusion contributions of the NH_4_V_4_O_10_@CNT fiber electrode at different scan rates. The formula is as follows [31,32]:*i(V)* = *k*_1_*v* + *k*_2_*v*^1/2^(1)

Here, *k_1_v* represents capacitive performance and *k*_2_*v*^1/2^ is diffusion-controlled. k_1_ and k_2_ are constants obtained from fitting curves of *iv*^1/2^ vs. k_1_*v*^1/2^. The result is shown in Figure 4e. The enclosed area means non-diffusion capacitive-controlled; according to calculations, the capacitance area accounted for 74.38% of the total CV area at the scan rate of 20 mV s^−1^. Furthermore, as the scan rate increased from 2 to 50 mV^−1^, the non-diffusion capacitance occupancy ratio distinctly increased from 54.98% to 85.77%, as Figure 4f shows, and further details are shown in Appendix A. This means the diffusion control of the fiber battery also occupied a large proportion at low scanning speed. The reason may be that NH_4_^+^ ions in the electrolyte can fully insert and extract with a nano-honeycomb structure at a low scan rate. Based on previous research on NH_4_^+^-ion batteries, it can be simply concluded that the reaction mechanisms are NH_4_^+^ ions de-intercalated and intercalated from the electrode. This corresponded to the NH_4_^+^ ion and O in the V-O layer being broken and re-bonded. The honeycomb-like nanostructure was conducive to NH_4_^+^-ion insertion and extraction; in this case, the NH_4_V_4_O_10_@CNT fiber battery had an excellent specific capacity.

PANI, one of the most commonly conductive polymer materials, is a potential anode material in the flexible energy storage field because of its advantages of low cost, simple processing, good conductivity, and high capacitance. Therefore, in this work, the PANI@CNT fiber electrode was used as an anode. The PANI@CNT fiber electrode was prepared by the electrochemical polymerization method with the three-electrode system. The characterization of PANI@CNT fiber is shown in Figure 5. SEM pictures revealed that PANI nanorods grew uniformly on fiber surfaces. A 50 nm diameter PANI nanorod was measured, and the diameter of the PANI@CNT fiber was 240 μm. The SEM images demonstrated that nanorods adhere to each other and form many holes, which was conducive to the electrochemical reaction between PANI nanorods and electrolyte ions on the electrode surface. The electrochemical properties of the PANI@CNT fiber were tested. The results showed better capacitance performance. The CV curve of the PANI@CNT fiber showed a very obvious oxidation peak (0.32 V vs. Ag/AgCl) and a reduction peak (0.08 V vs. Ag/AgCl) at a scan rate of 2 mV s^−1^ (Figure 5b). This means the redox of PANI was reversible under a potential window of −0.1 to 0.5 V. GCD curves of the PANI@CNT fiber also exhibited a high specific capacitance of 94.71 mAh cm^−3^ at a current of 0.08 mA, and the Coulombic efficiency could reach 93.5% (Figure 5d). Even when the current was increased to 0.2 mA, the Coulombic efficiency was still as high as 91.7%. The specific capacitances of PANI@CNT fiber at different densities were calculated (Appendix A). Meanwhile, CV curves all kept the same profile with different scan rates (Figure 5c), showing that the PANI@CNT fiber had good reversibility. This was consistent with the results of the GCD curves. In order to better understand the dynamics of charge transfer between electrodes and electrolytes, the EIS spectra of PANI@CNT fiber were also analyzed (Appendix A). The small Rct ensured fast and little-changed electron/ion transport in the fiber electrode.

In addition, the full cells were assembled with the PANI@CNT//NH_4_V_4_O_10_@CNT, and the electrode reaction process is depicted in Figure 6a. The ammonium ions were reversibly shuttled between the PANI anode and the NH_4_V_4_O_10_ cathode because the large interlayer space grew. The electrochemical performance was investigated, as shown in Figure 6b–g. The CV curves of the full cell in 1 mol L^−1^ (NH4)_2_SO_4_ aqueous solution with different scan rates from 1 mV s^−1^ to 50 mV s^−1^ are shown in Figure 6b. It can be observed that a pair of oxidization/reduction peaks appeared at around 0.42/0.18 V at a low scan rate, corresponding to reversible oxidation/reduction in the NH_4_V_4_O_10_ cathode, which was consistent with the voltage potential of the NH_4_V_4_O_10_@CNT fiber electrode in the half cell. Regarding the reaction mechanisms, it can be simply concluded that during the charge and discharge process, NH_4_^+^ was de-intercalated/intercalated from the electrode. The overall reactions were described by the formula (Appendix A). Furthermore, a low electrode charge transfer resistance (R_ct_) is demonstrated by the EIS profile of the full cells in Figure 6c. The bulk resistance is the intercept of the EIS at the real axis in the high frequency 44.9 Ω. Enlarging the EIS profile by the inset high–medium-frequency region, the R_ct_ was tested as just 1.8 Ω. The low Rct should be related to a fast interfacial electrode reaction between the electrode and electrolyte. Meanwhile, a slope of more than 45° can be observed in the low-frequency region, indicating that the reaction kinetics of the battery system was fast and that the battery system thereby had outstanding capability.

Attractively, the full cells showed a specific capacity of 7.27 mAh cm^−3^ at a current of 0.03 mA. With the current increased from 0.03 mA to 0.1 mA, the specific capacity was decreased due to the limitation of the electrode redox kinetic process. However, the discharge capacities of full cells were still 6.85, 6.4, 6.05, and 5.56 mAh cm^−3^ at currents of 0.04, 0.06, 0.08, and 0.1 mA, respectively (Figure 6d,e), verifying a good rate capability of the fiber battery. Meanwhile, the Coulombic efficiencies were increased to 78.27%, 86.54%, 93.01%, and 98.46% with the current densities of 0.02, 0.03, 0.04, and 0.05 mA cm^−2^, respectively (Figure 6f). The result was better than that of other NH_4_^+^-ion fiber batteries. It corresponded to a uniform honeycomb-like nanostructure where ion diffusion paths are shortened. Compared with previously reported fiber batteries, the specific capacity of the as-prepared cell was comparable or superior to that of other fibrous full cells (Appendix A), such as CC-CCH@CMO (3 mAh cm^−3^) [33], Ni–NiO fiber (237.8 uAh cm^−3^) [34], Co_3_O_4_/N-rGO (~0.5 mAh cm^−3^), and Co/Co-N-C (0.17 mAh cm^−3^) [35,36]. The long cycle-life of the full cells is depicted in Figure 6g. After 1000 cycles, a reversible specific capacity of 4.81 mAh cm^−3^ at the current of 0.5 mA was still maintained with a capacity retention of 72.1% of the initial capacity, denoting good reversibility and cyclic stability of the fiber batteries. Furthermore, the Coulombic efficiencies were almost over 80% during the whole 1000-cycle process, consistent with the result as Figure 6f showed. Coulomb efficiency is low at the beginning and gradually stabilizes above 80% after the electrode is activated, which may be because the electrode structure is relatively stable [37]_._ The defects of NH_4_V_4_O_10_@CNT fiber cathode may be heteroatom doping defects causing complicated side reactions. Nitrogen atoms appeared as heteroatoms when the active materials are grown in a hydrothermal reaction. Nitrogen doping defects may improve the rate of ion–electron conduction. However, the reaction is irreversible, which causes poor cycle stability.

Flexibility is another important indicator of fiber full cells for wearable energy practical applications. The fiber battery exhibits an excellent specific capacity of 6.48 mAh cm^−3^ under different bending states (Appendix A). Moreover, the full cells can retain 90% of their initial capacity after 500 cycles at an angle of 180°; thus, they exhibited excellent bending stability and have great potential as a flexible battery to be applied in digital smart textiles (Appendix A).

## 4. Conclusions

In summary, an ammonium-ion fiber battery has been developed with a honeycomb-like NH_4_V_4_O_10_@CNT cathode in this work. The honeycomb-like nanostructure and large interlayer spacing were of benefit to the NH_4_^+^-ion diffusion, endowing the battery with high energy storage performance and excellent flexibility. The as-prepared NH_4_V_4_O_10_@CNT cathode delivered a reversible specific capacity of 241.06 mAh cm^−3^ at a current of 0.2 mA with a high Coulombic efficiency of 97.3%. In addition, a high-strength NH_4_V_4_O_10_@CNT fiber battery was also confirmed. The NH_4_^+^-ion fiber full cell achieves a reversible capacity of 6.86 mAh cm^−3^ at 0.04 mA. Moreover, the fiber battery demonstrates good flexibility and retains high capacitance retention in a bending state, implying a potential application in the smart wearable textile field.

## Figures and Tables

**Figure 1 polymers-14-04149-f001:**
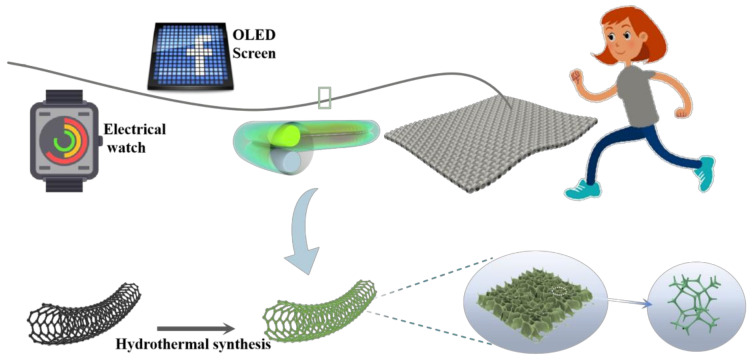
Schematic illustration of the NH_4_V_4_O_10_@CNT electrode and the flexible energy storage application.

**Figure 2 polymers-14-04149-f002:**
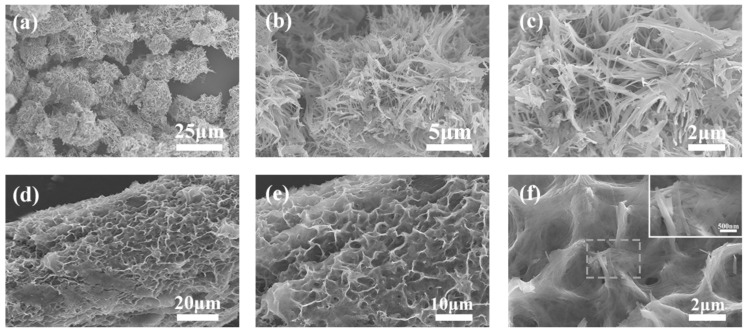
Scanning electron microscope (SEM) images; (**a**–**c**) the SEM images of NH_4_V_4_O_10_ powder with different magnification; (**d**–**f**) the SEM images of NH_4_V_4_O_10_@CNT fiber with different magnification.

**Figure 3 polymers-14-04149-f003:**
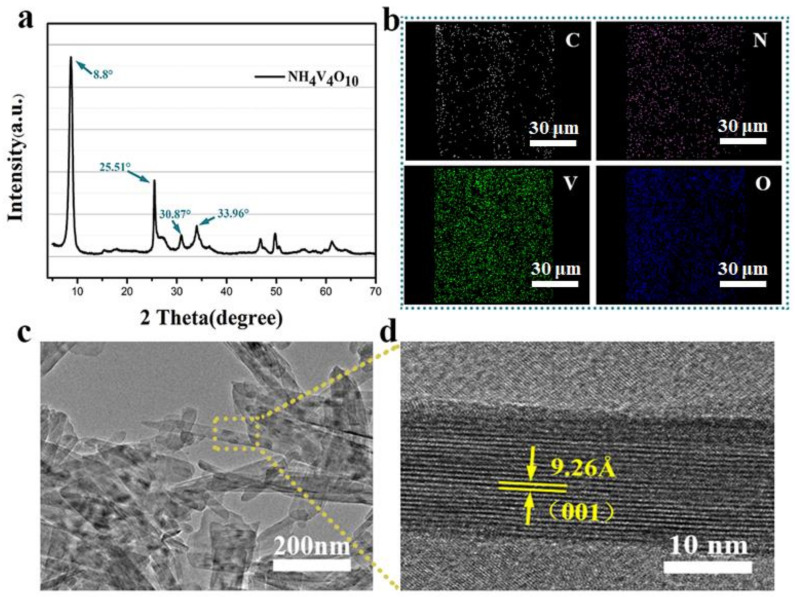
(**a**) X-ray diffraction (XRD) pattern; (**b**) EDS mapping of C, N, V, and O; (**c**) transmission electron microscopy (TEM) images; (**d**) HRTEM images of NH_4_V_4_O_10_ at different magnifications.

**Figure 4 polymers-14-04149-f004:**
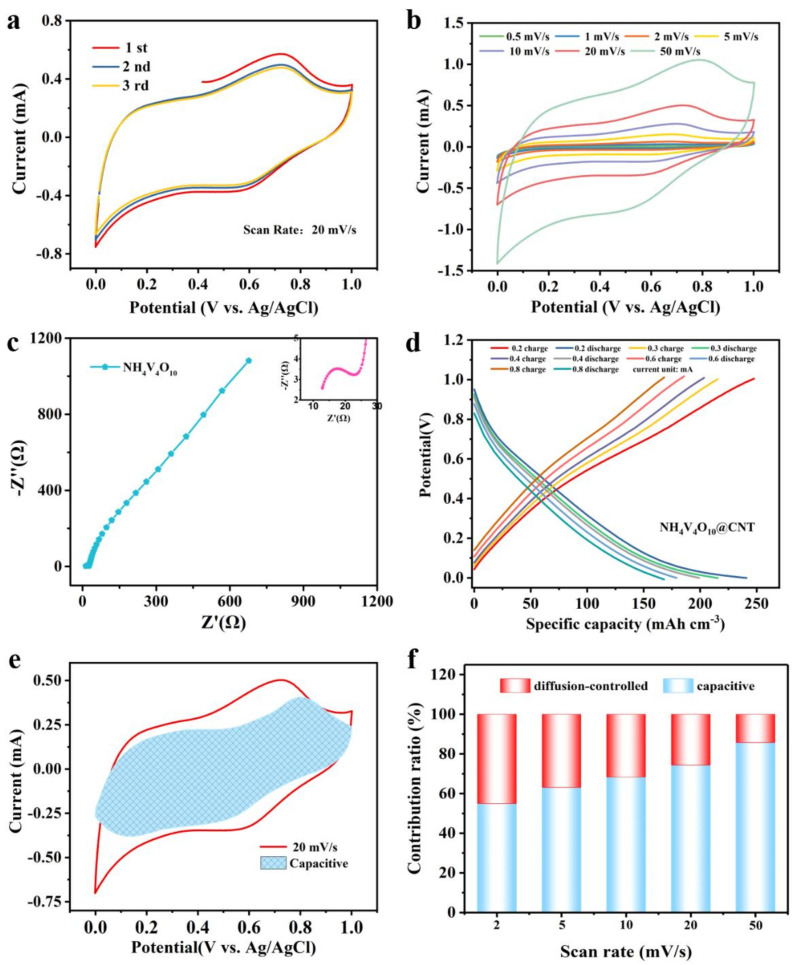
Electrochemical performance of NH_4_V_4_O_10_@CNT fiber with 1 mol L^−1^ (NH_4_)_2_SO_4_ electrolyte in a three-electrode system: (**a**) the CV test of NH_4_V_4_O_10_@CNT fiber at a scan rate of 20 mV s^−1^; (**b**) the CV test of NH_4_V_4_O_10_@CNT fiber at different scan rates; (**c**) the EIS curve of NH_4_V_4_O_10_@CNT fiber; (**d**) galvanostatic charge/discharge measurement of NH_4_V_4_O_10_@CNT fiber at different current densities; (**e**) CV curves with the capacitive fraction shown by the shaded area at a scan rate of 20 mV s^−1^; (**f**) electrode dynamics of NH_4_V_4_O_10_@CNT fiber with the percent of pseudocapacitive contribution at different scan rates.

**Figure 5 polymers-14-04149-f005:**
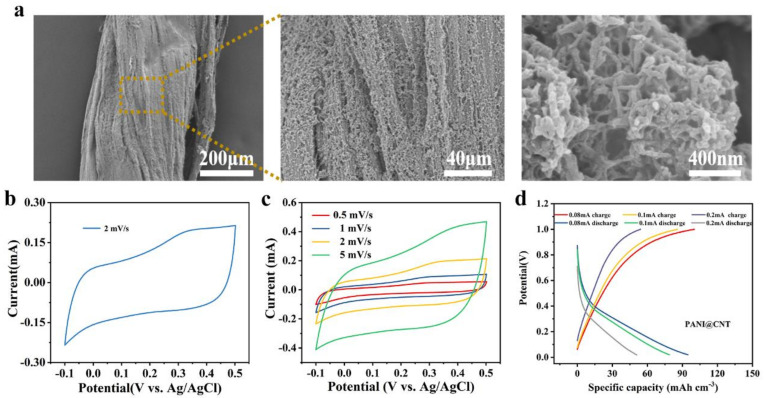
Characterization of the PANI@CNT fiber: (**a**) the morphology of PANI at different magnifications; (**b**) CV curve of PANI@CNT fiber at a scan rate of 2 mV s^−1^; (**c**) CV curves of PANI@CNT fiber at different scan rates; (**d**) GCD curves of PANI@CNT fiber at different current densities.

**Figure 6 polymers-14-04149-f006:**
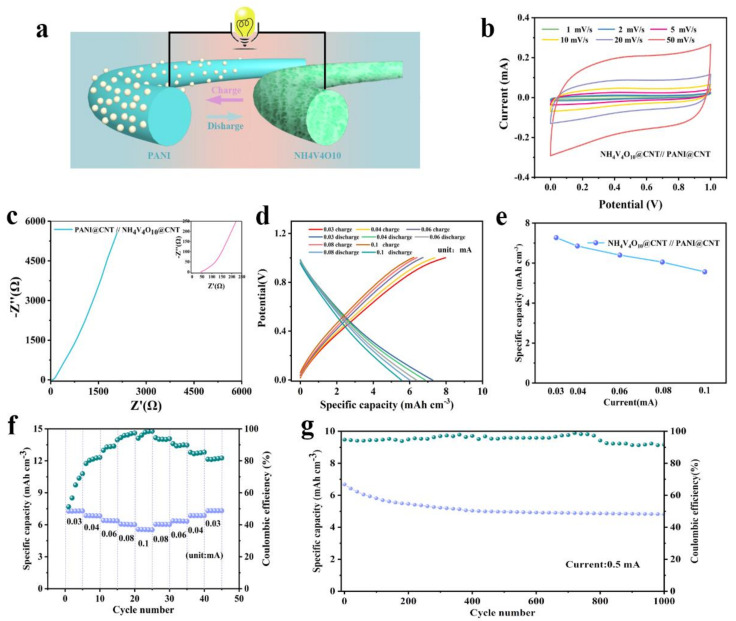
Electrochemistry properties of the aqueous NH_4_V_4_O_10_@CNT//PANI@CNT full cell. (**a**) Schematic illustration of the fiber battery full cell. (**b**) CV curves of the full cell at different scan rates. (**c**) EIS profile and inset EIS curve with the high-frequency region. (**d**) Galvanostatic charge/discharge curves of NH_4_V_4_O_10_@CNT//PANI@CNT fibers full cell at different current densities. (**e**) Specific capacity at different current densities. (**f**) Rate performance of the full cell at different currents from 0.03 to 0.1 mA. (**g**) Long-term cycle performance at a current of 0.5 mA.

## Data Availability

Not applicable.

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
