# Peer review of "A Honeycomb-like Ammonium-Ion Fiber Battery with High and Stable Performance for Wearable Energy Storage"

_polymers, 2022, doi:10.3390/polym14194149_

Round 1

Reviewer 1 Report

In this manuscript, the authors reported a flexible fibre battery consisting of an NH4V4O10@CNT cathode and a PANI@CNT anode. By anchoring NH4V4O10 nanorods on carbon nanotubes, the electrode shows promoted ion transfer and improved reaction rate. The performance is attractive, and the manuscript is well presented. Overall, I would like to suggest a major revision for this manuscript. Here are some comments that could help improve the quality of this manuscript.

1)“Redox includes reduction and oxidization. In the manuscript, the authors use redox/reducein many places. Can the authors discuss or explain the difference between these?

2) It is well studied that defects in the NH4V4O10 introduced by the hydrothermal reaction can significantly affect the performance of the cathode. Can the authors discuss this in the manuscript? Are there any defects in the cathode? If so, what kind of defects are there and how will they affect the performance?

3)The scale bar of Figure 3b should be provided in the main text.

4)In the full cell, the specific capacity is drastically decreased. How to calculate the specific capacity should be discussed. Does it include all electrode and battery packages?

5) what’s the weight percentage of NH4V4O10 and PANI in the electrodes? How does it affect the capacitance and intercalation capacity ratio?

6)what’s the negative/positive capacity ratio (N/P ratio)?

Author Response

Thank you very much for the time and effort to have our manuscript reviewed, and for giving us the opportunity to revise our manuscript. In summary, we have fully revised the manuscript according to the reviewers’ comments. For your convenience, the main revisions are marked in red. Thanks a lot for your consideration.

Reviewer 2 Report

1) Current densities should have the units of current/area, as in A/cm^2, please correct all the units in the main text and figure captions;

2) At Line 82 on Page 2, it should be "NH4-ion", not "NH-ion";

3) Please rewrite the caption of Figure 1, I don't see any full-cell in the figure;

4) Why is 12 ohm calculated from Figure 4(c) a small resistance, compared to what value? Please reference other works that have higher charge-transfer resistances calculated from the same method;

5) In Figure 6(f), coulombic efficiencies are quite low (<60%) at small current densities (0.03 mA and 0.04 mA), why? Also please explain the trend of Coulombic efficiencies increasing when one increases the current density from 0.03 mA to 0.1 mA. 

Author Response

(The authors gave the same response as above.)

Round 2

Reviewer 1 Report

The authors have addressed all of my concerns. I would like to suggest publishing it on Polymers.

Reviewer 2 Report

I believe the manuscript has been sufficiently improved to warrant publication. Thanks.